# DVIB: Towards Robust Multimodal Recommender Systems via Variational Information Bottleneck Distillation

## Abstract

In multimodal recommender systems (MRS), integrating various modalities helps to model user preferences and item characteristics more accurately, thereby assisting users in discovering items that match their interests. Although the introduction of multimodal information offers opportunities for performance improvement, it will increase the risks of inherent noise and information redundancy, posing challenges to the robustness of MRS. Many existing methods typically address these two issues separately either by introducing perturbations at the model input for robust training to handle noise or by designing complex network structures to filter out redundant information. In contrast, we propose the DVIB framework to simultaneously address both issues in a simple manner. We found that moving the perturbations from the input layer to the hidden layer, combined with feature self-distillation, can mitigate noise and handle information redundancy without altering the original network architecture. Additionally, we also provide theoretical evidence for the effectiveness of DVIB, demonstrating that the framework not only explicitly enhances the robustness of model training but also implicitly exhibits an information bottleneck effect, which effectively reduces redundant information during multimodal fusion and improves feature extraction quality. Extensive experiments show that DVIB consistently improves the performance of MRS across different datasets and model settings, and it can complement existing robust training methods, representing a promising new paradigm in MRS. The code and all models will be released online.

## CCS Concepts

• **Information systems** → **Recommender systems**; • **Computing methodologies** → **Knowledge representation and reasoning**; **Artificial intelligence**.

## Keywords

Multimodal Recommender System, Robust Training, Variational Information Bottleneck, Feature Distillation

## 1 Introduction

**Relevance to the Web and to the track.** Recommender systems (RS) are essential for guiding users through the overwhelming variety of options on the web, uncovering tailored items and services [67]. In recent years, deep learning-based approaches [7, 12, 22, 51, 63, 69] have become prevalent in these systems, using historical interactions to predict preferences and enhance personalization. With the rise of diverse data like text, images, and videos [16, 18, 66], multimodal recommender systems (MRS) [42, 73] have emerged to tackle issues like data scarcity and cold start challenges [75].

Although the introduction of multimodal data provides more dimensions of user information for RS, enhancing the diversity

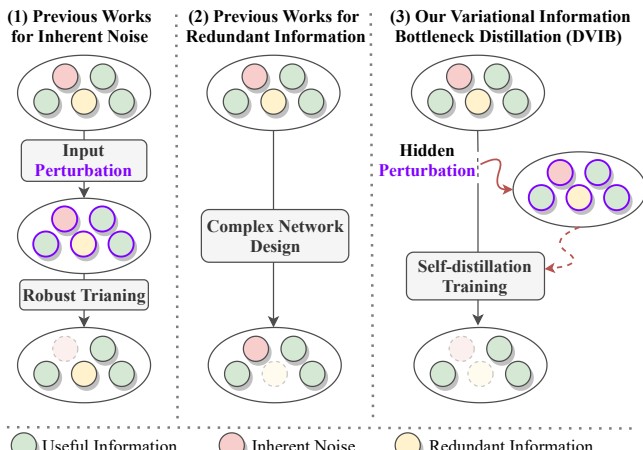

Figure 1: The potential risks of MRS and their solutions. (1) For inherent noise, it is common to add perturbations to the model's input and mitigate the impact of noise through robust training. (2) For redundant information, it can often be reduced by designing complex network architectures. (3) Our proposed DVIB framework shifts the perturbation from the input layer to the hidden layer, combined with self-distillation. Both Section 3.1 and Section 4 demonstrate, through theoretical and experimental results, that the proposed framework can mitigate both risks due to the implicit effect of Variational Information Bottleneck (VIB), improving model performance without any additional computational cost during inference.

and accuracy of recommendations, it simultaneously increases **two potential risks** that affect the robustness of the RS.

**(1) The first risk is the inherent noise**. For instance, on e-commerce platforms, merchants or users may upload some blurry item images or provide textual information, like descriptions and comments, that do not completely match the items, introducing inherent harmful noise to the model [64, 67]. When multimodal data, such as images and text, are integrated, the risk of models being further contaminated by harmful noise increases, as different modalities carry noise. This limits the performance gains brought by multimodal fusion. There are a variety of methods [3, 9, 11, 35, 40, 46, 58, 59, 62] to mitigate inherent noise, as shown in Fig. 1 (a) by introducing input perturbation into MRS and conducting robust training to improve the robustness of MRS.

**(2) The second risk is redundant information**. Although multimodal data provide the system with rich complementary information, they also contain a large amount of irrelevant or duplicate features [6, 31, 32, 36]. For example, an image of a coat for sale may include the background from the model's photoshoot, while the item description might mention "free delivery," which is unrelated to the item itself. When multiple modalities are integrated,

redundant information increases the difficulty for the model to distinguish effective features within the multimodal data. This interferes with the model's learning process and reduces its focus on relevant features. To address redundant information, as shown in Fig. 1 (b), Many existing MRS systems [31, 33, 36, 49, 52, 54] extract better features and mitigate information redundancy by proposing sophisticated model designs.

For this requirement, in this paper, we propose a highly simple-yet-effective framework, **Variational Information Bottleneck Distillation (DVIB)**, which can **simultaneously** handle both **noise** and **redundant information risks** without any additional computational cost during inference. As shown in Fig. 1 (c) and Section 3, DVIB framework shifts the perturbation from the input layer to the hidden layer, combined with self-distillation. On one hand, such robust training can explicitly mitigate the impact of inherent noise. On the other hand, we theoretically demonstrate that DVIB implicitly possesses the effects of VIB (see Section 3.1). This means that the DVIB framework can effectively extract task-relevant key information and filter out redundant information, thereby helping to improve MRS performance. In Section 4 and 5, we also conduct extensive experiments demonstrating performance improvements in various multimodal recommendation systems across different datasets, validating the effectiveness and compatibility of our framework. Our contributions are summarized as follows:

- We propose the simple-yet-effective DVIB framework, which can mitigate the inherent noise and redundant information risks in MRS simultaneously without altering the model architecture, thereby substantially boosting the robustness of RS.
- We also provide strong theoretical evidence for the relationship between our framework and variational information bottleneck, which theoretically supports the effectiveness of the proposed DVIB.
- Extensive experiments show that DVIB consistently enhances the MRS performance across different datasets and model settings, and it's compatible with some existing robust training methods.

## 2 Preliminary

**Multimodal Recommender Systems (MRS).** We define a set of users $\mathcal{U} = \{u_1, u_2, ..., u_{|\mathcal{U}|}\}$ and a set of items $\mathcal{I} = \{i_1, i_2, ..., i_{|\mathcal{I}|}\}$. Each user $u \in \mathcal{U}$ has a subset of items $\mathcal{I}u \subseteq \mathcal{I}$ for which they have shown positive feedback. Each item $i \in \mathcal{I}$ is characterized by visual features $v_i \in \mathcal{V}$ and textual features $t_i \in \mathcal{T}$ in this paper. The multimodal recommendation model $\mathbf{MRS}(\cdot)$ computes the user-item preference score $y_{u,i}$ as follows:

$$y_{u,i} = \mathbf{MRS}(u, i, v_i, t_i, \mathcal{I}_u \mid \Theta), \qquad (1)$$

where $\Theta$ are the model parameters. The score $y_{u,i}$ indicates the likelihood of recommending item $i$ to user $u$, with higher scores suggesting a better match.

Given our goal to propose a universal robust enhancement framework for MRS, considering the diverse model structures of MRS, we need to first formulate MRS from a more high-level perspective, as illustrated by the black line in Fig. 2. Let's denote the input from different modalities as $M_i$, where , $i = 1, 2, .., n$. $Y$ is the label of training data and both $H$ and $Z$ are the hidden feature of the MRS. Subsequently, MRS first undergo two stages: feature fusion and

feature extraction, i.e.,

$$H = f_{\theta_1}[\text{Fusion}(M_1, M_2, ..., M_n)], \quad Z = f_{\theta_2}(H), \qquad (2)$$

Following the output stage, the loss function is constructed using Eq. (3). Depending on the various MRS designs, $\mathcal{L}_0$ can take on various forms, such as the Bayesian personalized ranking loss [15, 41, 72], or other supplementary losses [48, 79] to enhance model performance. In this paper, we uniformly denote the original general loss function as $\mathcal{L}_0$.

$$\mathcal{L}_0 \equiv \mathcal{L}_0(f_{\theta_3}(Z), Y). \qquad (3)$$

**The Bounds in the VIB.** In MRS training, when mapping between training data $X \rightarrow Y$, the goal of the VIB is to optimize the neural network with parameters $\theta$ to impose constraints on the hidden layer $H$, aiming to improve the robustness of the model [1]. Specifically, the optimization objective of the VIB is to maximize

$$I(H, Y; \theta) - \beta \cdot I(H, X; \theta), \qquad (4)$$

where $\beta$ is a constant weight and $I(A, B; \theta)$ represents the mutual information between $A$ and $B$. In Eq. (4), $I(H, Y; \theta)$ encourages $H$ to contain more information useful for predicting label $Y$, while $I(H, X; \theta)$ encourages $H$ to "ignore" information about the input feature $X$. A hidden feature $H$ that satisfies both is considered a good feature. Since $H$ can ignore irrelevant details in $X$ that are unrelated to the target $Y$, $H$ becomes robust to noise. Additionally, maximizing the compression of $H$ with respect to $X$, it helps mitigate information redundancy from $X$. The first commonly used bound [1] in Eq. (4) is

$$I(H, Y) \geq \int dX dY dH p(X) p(Y|X) p(H|X) \log p(Y|H), \quad (5)$$

and for the other term $I(H, X)$, we have the following bound

$$I(H, X) \leq \int dH dX p(X) p(H|X) \log[p(H|X)/r(H)], \qquad (6)$$

where $r(H)$ is a given prior distribution.

## 3 Methodology

In this section, we detail our proposed DVIB framework in Section 3.1, which is a universal method for enhancing the MRS robustness, and it requires no extra computational cost during the model inference. Subsequently, we present theoretical support for DVIB's efficacy in Section 3.2, showing it not only explicitly enhances robust training but also implicitly exhibits an information bottleneck effect, which effectively reduces redundant information and noise during multimodal fusion and improves feature extraction quality.

## 3.1 Variational Information Bottleneck Distillation (DVIB)

Unlike many previous models [3, 9, 46, 59, 62] that perform robust training at the input layer, our proposed DVIB shifts the perturbations from the input layer to the hidden layer. Combined with feature self-distillation, we find that this simple training approach, due to its implicit VIB effect (refer to section 3.2), can effectively mitigate the risks of inherent noise and information redundancy simultaneously, without altering the original network architecture.

Figure 2: An overview of the DVIB. The MRS with $n$ modalities $\{M_i\}_{i=1}^{n}$ typically consists of three stages: feature fusion, feature extraction, and feature output. The black lines represent the forward path of the original (Org) MRS. The red lines indicate the additional algorithms required by DVIB. ① and ② are the general loss of Org path and extra DVIB path in Eq. (3) and Eq. (9); ③ and ④ are self-distillation loss in Eq. (10) and regularization term in Eq. (13), respectively.

Specifically, as indicated by the red line in Fig. 2, we build upon the Org path, i.e. the process of the original MRS, which is based on Eq. (2) and Eq. (3), by adding some additional steps, including

$$Z_\delta = f_{\theta_2}(H + \delta H), \tag{7}$$

where $\delta H$ represents the perturbations. $f_{\theta_i}, i = 1, 2, 3$ are shared in Org path and extra DVIB path. Additionally, we introduce three loss functions, i.e.,

$$\mathcal{L}^{\text{DVIB}} = \mathcal{L}_0^{\text{DVIB}} + \mathcal{L}_d^{\text{DVIB}} + \mathcal{L}_r^{\text{DVIB}}, \tag{8}$$

where $\mathcal{L}_0^{\text{DVIB}}, \mathcal{L}_d^{\text{DVIB}}, \mathcal{L}_r^{\text{DVIB}}$ are general loss, self-distillation loss, and regularization term, respectively. The first, similar to Eq. (3), requires that even with the newly added perturbations, the forward feature should also be as close to the label as possible. Therefore, we have

$$\mathcal{L}_0^{\text{DVIB}} \equiv \gamma_0 \cdot \mathcal{L}_0(f_{\theta_3}(Z_\delta), Y), \tag{9}$$

where $\gamma_0 \in \mathbb{R}$ is a weight. Next, we distill the features obtained through robust training with added perturbations into the Org path using self-distillation loss $\mathcal{L}_d^{\text{DVIB}}$. In DVIB, we design the extra DVIB path shown in Fig. 2 to not participate in the model's inference phase after training. Therefore, we transfer the high-quality features learned by $Z_d$ to $Z$ through self-distillation. Specifically, the self-distillation loss is

$$\mathcal{L}_d^{\text{DVIB}} \equiv \gamma_d \cdot \text{JS}(Z_\delta || Z). \tag{10}$$

where JS is JS-divergence [37] and $\gamma_d$ is a consistency weight.

Next, we consider the details of the regularization term $\mathcal{L}_r^{\text{DVIB}}$, and in this paper, $\mathcal{L}_r^{\text{DVIB}}$ is designed by the perturbations $\delta H$. Specifically, depending on the method of perturbations, DVIB can primarily take two forms. One is perturbations [39] constructed through constant gaussian noise

$$\delta H \sim \mathcal{N}(\delta H; \mathbf{0}, \sigma^2 \mathbf{I}), \tag{11}$$

which we refer to as $\text{DVIB}_c$ and $\sigma$ is a constant. The other is perturbations that self-adjust adaptively, i.e.,

$$\delta H = \phi \cdot \epsilon, \quad \epsilon \sim \mathcal{N}(\epsilon; \mathbf{0}, \mathbf{I}), \tag{12}$$

which we call $\text{DVIB}_a$. The $\phi \in \mathbb{R}$ is a learnable scale. For the two distinct $\delta H$ shown in Eq. (11) and Eq. (12), their respective

regularization terms $\mathcal{L}_r^{\text{DVIB}}$ can be defined as

$$\mathcal{L}_r^{\text{DVIB}} = \begin{cases} \gamma_r \left[ \alpha_0 H^\top H + \beta_0 \left( \frac{\phi}{a} - \log \frac{\phi}{a} - 1 \right) \right], & \text{if DVIB}_a \\ 0, & \text{if DVIB}_c \end{cases}, \tag{13}$$

where $\gamma_r, \alpha_0, \beta_0$ and $a$ are hyperparameters. Here, we use a 3-layers neural network of $H$ to measure the scale $\phi$, i.e.,

$$\phi = \text{Net}_\phi(H) = \sigma_1(\mathbf{W}_1(\sigma_2(\mathbf{W}_2(\sigma_3(\mathbf{W}_3 H))))), \tag{14}$$

where $\sigma_1$ is Sigmoid function and $\sigma_2, \sigma_3$ are ReLU. $\mathbf{W}_1 \in \mathbb{R}^{d_1 \times 1}, \mathbf{W}_2 \in \mathbb{R}^{d_2 \times d_1}$, and $\mathbf{W}_3 \in \mathbb{R}^{d_3 \times d_2}$ are learnable matrices. In summary, the complete process of DVIB is shown in Alg. 1.

---

**Algorithm 1** The details of DVIB framework

---

**Input:** As shown in Fig. 2, the input feature $X$ and label $Y$. The hidden features $H$ and $Z$. The learnable parameters $\Theta = [\theta_1, \theta_2, \theta_3]$. The hyperparameters $\sigma, \alpha_0, \beta_0, a$. The network $\text{Net}_\phi(\cdot)$.
**Output:** Model parameters $\Theta$.

1: Construct loss $\mathcal{L}_0$ in Eq. (3) by Eq. (2)      ▷ Original Training
2: **if** $\text{DVIB}_c$ **do**      ▷ DVIB Training
3:      $\delta H \sim \mathcal{N}(\delta H; \mathbf{0}, \sigma^2 \mathbf{I})$
4:      Construct regularization terms $\mathcal{L}_r^{\text{DVIB}} = 0$
5: **if** $\text{DVIB}_a$ **do**
6:      $\delta H = \phi \cdot \epsilon = \text{Net}_\phi(H) \cdot \epsilon, \quad \epsilon \sim \mathcal{N}(\epsilon; \mathbf{0}, \mathbf{I})$
7:      Construct $\mathcal{L}_r^{\text{DVIB}} = \gamma_r [\alpha_0 H^\top H + \beta_0 \left( \frac{\phi}{a} - \log \frac{\phi}{a} - 1 \right)]$
8: Construct general loss $\mathcal{L}_0^{\text{DVIB}}$ in Eq. (9)
9: Construct self-distillation loss $\mathcal{L}_d^{\text{DVIB}}$ in Eq. (10)
10: Let $\mathcal{L}^{\text{DVIB}} = \mathcal{L}_0^{\text{DVIB}} + \mathcal{L}_d^{\text{DVIB}} + \mathcal{L}_r^{\text{DVIB}}$
11: Optimize $\Theta$ and $\phi$ to minimize $\mathcal{L}^{\text{DVIB}} + \mathcal{L}_0$
12: **return** $\Theta$

---

## 3.2 Theoretical Insight of DVIB

In this section, we present theoretical evidence that both of our proposed DVIB models, due to its implicit VIB effect, can simultaneously mitigate inherent noise and information redundancy. We will demonstrate this in three steps. First, we show that the regularization term $\mathcal{L}_r^{\text{DVIB}}$ in the case of $\text{DVIB}_a$ corresponds to imposing a specific distributional constraint. Next, we prove that $\text{DVIB}_c$ is a special case of $\text{DVIB}_a$. Finally, using the conclusions from the previous proofs, we analyze the effectiveness of the DVIB framework in addressing information redundancy and noise in a unified manner. We use notation $A \propto B$ to indicate that $A$ is approximately equivalent to $B$.

THEOREM 3.1. *For $\text{DVIB}_a$, minimizing $\mathcal{L}_r^{\text{DVIB}} = \alpha_0 \cdot H^\top H + \beta_0 \cdot \left( \frac{\phi}{a} - \log \frac{\phi}{a} - 1 \right)$ is approximately equivalent to minimize the KL-divergence between $\mathcal{N}(H, \phi \mathbf{I})$ and $\mathcal{N}(\mathbf{0}, a\mathbf{I})$.*

PROOF. Consider $H$ is a $d$−dimensional vector, we have

$$\min \mathcal{L}_r^{\text{DVIB}} = \min \alpha_0 \cdot H^\top H + \beta_0 \cdot \left( \frac{\phi}{a} - \log \frac{\phi}{a} - 1 \right) \tag{15}$$

$$\propto \min \alpha_0 \|H\|_2^2 + \beta_0 (d\frac{\phi}{a} - \log(\frac{\phi}{a})^d - d).$$

Let $\Sigma_\phi = \phi \mathbf{I}$ and $\Sigma_a = a\mathbf{I}$, where $\mathbf{I}$ is identity matrix, we have

$$d\frac{\phi}{a} - \log(\frac{\phi}{a})^d - d = \text{tr}(\Sigma_a^{-1} \Sigma_\phi) - \log \det(\Sigma_a^{-1} \Sigma_\phi) - d. \tag{16}$$

Therefore, for $\mathcal{L}_r^{\text{DVIB}}$, we have

$$\min \mathcal{L}_r^{\text{DVIB}} \propto \min \alpha_0 \|H\|_2^2 + \beta_0 (d\frac{\phi}{a} - \log(\frac{\phi}{a})^d - d).$$
$$\propto \min H^\top \Sigma_a^{-1} H + \beta_0 (\text{tr}(\Sigma_a^{-1}\Sigma_\phi) - \log \det(\Sigma_a^{-1}\Sigma_\phi) - d).$$
$$\propto \min \text{KL}(\mathcal{N}(H, \phi\mathbf{I}) || \mathcal{N}(0, a\mathbf{I})). \tag{17}$$

The final step of the above formula is due to the fact that $H^\top \Sigma_a^{-1} H + \beta_0 (\text{tr}(\Sigma_a^{-1}\Sigma_\phi) - \log \det(\Sigma_a^{-1}\Sigma_\phi) - d)$ matches the form of the KL-divergence formula between two gaussian distribution.  $\square$

> **Theorem 3.2.** *As $\alpha_0 \to 0$ and $a = \sigma^2$, $DVIB_a$ tends towards $DVIB_c$ in the sense of minimizing the regularization term $\mathcal{L}_r^{DVIB} = \alpha_0 \cdot H^\top H + \beta_0 \cdot \left(\frac{\phi}{a} - \log \frac{\phi}{a} - 1\right)$.*

**Proof.** Given the function $g(x) = x - \log x - 1$, it is noted that $x = 1$ is the only real solution to $g(x)' = 0$. By analysis, it is known that $g(x)$ is monotonically decreasing when $x \in (0, 1)$ and monotonically increasing when $x \in [1, \infty)$, meaning $g(x)$ achieves its minimum value at $x = 1$.

Therefore, as $\alpha_0 \to 0$ and $a = \sigma^2$, we have:

$$\alpha_0 \cdot H^\top H + \beta_0 \cdot \left(\frac{\phi}{a} - \log \frac{\phi}{a} - 1\right) \to \beta_0 \cdot \left(\frac{\phi}{\sigma^2} - \log \frac{\phi}{\sigma^2} - 1\right) \tag{18}$$

At this point, the minimum of $\min \mathcal{L}_r^{\text{DVIB}}$ is achieved when $\phi/\sigma^2 = 1$, which implies $\phi = \sigma^2$. At this time, Eq. (11) and Eq. (12) are equivalent.  $\square$

> **Theorem 3.3.** *minimizing $\mathcal{L}_0^{DVIB} + \mathcal{L}_r^{DVIB}$ is approximately equivalent to maximize the mutual information $I(H, Y)$ and minimize the mutual information $I(H, X)$.*

**Proof.** As described in Section 2, the specific design of $\mathcal{L}_0 \equiv \mathcal{L}_0(f_{\theta_3}(Z), Y)$ depends on different MRS algorithms, but generally, they all aim for $f_{\theta_3}(Z)$ to be as close to the label $Y$ as possible. Therefore, we can assume

$$\mathcal{L}_0 \equiv \mathcal{L}_0(f_{\theta_3}(Z), Y) \propto -\log p(Y|f_{\theta_3}(Z)). \tag{19}$$

Additionally, according to Theorem 3.2, the regularization terms of $\text{DVIB}_a$ and $\text{DVIB}_c$ can be uniformly considered in the form of $\mathcal{L}_r^{\text{DVIB}} = \alpha_0 \cdot H^\top H + \beta_0 \cdot \left(\frac{\phi}{a} - \log \frac{\phi}{a} - 1\right)$. Hence, for $\mathcal{L}_0^{\text{DVIB}} + \mathcal{L}_r^{\text{DVIB}}$, according to Theorem 3.1, we have

$$\mathcal{L}_0^{\text{DVIB}} + \mathcal{L}_r^{\text{DVIB}} \propto \mathcal{L}_0(f_{\theta_3}(Z_\delta), Y) + \text{KL}(\mathcal{N}(H, \phi\mathbf{I}) || \mathcal{N}(0, a\mathbf{I}))$$
$$= \mathcal{L}_0(f_{\theta_3}(f_{\theta_2}(H + \delta H)), Y)$$
$$+ \text{KL}(\mathcal{N}(H, \phi\mathbf{I}) || \mathcal{N}(0, a\mathbf{I})). \tag{20}$$

From Eq. (19), $\mathcal{L}_0(f_{\theta_3}(f_{\theta_2}(H+\delta H)), Y) \approx -\log p(Y|H)$. Let $p(H|x) = \mathcal{N}(H, \phi\mathbf{I})$ and $r(H) = \mathcal{N}(0, a\mathbf{I})$. Therefore, Eq. (20) is equivalent to optimizing follwoing objective function

$$\mathbb{E}_{(X,Y) \sim p(X,Y)} \{\mathbb{E}_{H \sim p(H|X)}[-\log p(Y|H)]\}$$
$$+ \mathbb{E}_{(X) \sim p(X)} \beta_0 \text{KL}(p(H|X) || r(H)). \tag{21}$$

Let $T_1 = \mathbb{E}_{(X,Y) \sim p(X,Y)} \{\mathbb{E}_{H \sim p(H|X)}[-\log p(Y|H)]\}$ and we set that $T_2 = \mathbb{E}_{X \sim p(X)} \{\text{KL}(p(H|X) || r(H))\}$. From Eq. (6) we have

$$T_2 = \mathbb{E}_{X \sim p(X)} \text{KL}(p(H|X) || r(H))$$
$$= \mathbb{E}_{X \sim p(X)} \int p(H|X) \log(p(H|X)/r(H)) dH.$$
$$= \int dH dX p(X) p(H|X) \log(p(H|X)/r(H)) \geq I(H, X). \tag{22}$$

Moreover, for $T_1$, according to Eq. (5), we have

$$T_1 = \mathbb{E}_{(X,Y) \sim p(X,Y)} \{\mathbb{E}_{H \sim p(H|X)}[-\log p(Y|H)]\}$$
$$= -\mathbb{E}_{(X,Y) \sim p(X,Y)} \left\{\int dH p(H|X) \log p(Y|H)\right\}$$
$$= -\int dX dY dH p(x) p(Y|X) p(H|X) \log p(Y|H) \geq -I(H, Y). \tag{23}$$

$\square$

Thus, according to Theorem 3.3, minimizing the loss function $\beta_0 T_2 + T_1$ helps to minimize the mutual information $\beta_0 I(H, X) - I(H, Y)$. At this point, the mutual information between the hidden feature $H$ and label $Y$ will be as large as possible, while the mutual information between $H$ and input feature $X$ will be as small as possible. This aligns with the VIB effect mentioned in Section 2, so our proposed DVIB framework implicitly reduces redundant information and noise during multimodal fusion and enhances feature extraction quality. The step of adding perturbations in the hidden layer for robust training explicitly increases the model's ability to resist noise. Meanwhile, the high-quality features obtained through these effects are distilled to the original network's forward features $Z$ by the self-distillation loss $\mathcal{L}_d^{\text{DVIB}}$, thereby improving the performance of the MRS.

So far, we have demonstrated the advantages of DVIB, which is simple, theoretically well-founded, and highly versatile. In the following Section, we will further illustrate DVIB's ability to enhance the performance of various MRS from an experimental perspective, as well as its compatibility with some existing robustness enhancement methods.

## 4 Experiments

### 4.1 Experimental Settings

**Datasets.** We employ three widely-used multimodal datasets from the Amazon Review Data [28], including Baby, Sports and Clothing. These datasets consist of both textual and visual features of items, see Appendix for details. For consistency and rigor in feature extraction, we follow the established preprocessing procedure outlined in MMRec [74]. Furthermore, we use the Pinterest dataset [13] to assess the compatibility of DVIB with some existing robust training methods [46, 67].

**Metrics.** To evaluate the performance of MRS, we emphasize Top-5 accuracy since recommendations in the highest-ranking positions hold greater significance in practical applications [47]. We employ four widely adopted metrics [14, 45, 61, 73]: Recall (REC), Normalized Discounted Cumulative Gain (NDCG), Precision (PREC), and Mean Average Precision (MAP). These metrics can provide a comprehensive evaluation by focusing on different aspects of the

| Models | Baby | | | | Sports | | | | Clothing | | | |
|---|---|---|---|---|---|---|---|---|---|---|---|---|
| | REC | NDCG | PREC | MAP | REC | NDCG | PREC | MAP | REC | NDCG | PREC | MAP |
| VBPR [15] | 0.0271 | 0.0177 | 0.0061 | 0.0141 | 0.0365 | 0.0243 | 0.0080 | 0.0197 | 0.0193 | 0.0131 | 0.004 | 0.0109 |
| VBPR+DVIB$_c$ | 0.0281 | 0.0185 | 0.0064 | 0.0153 | 0.0383 | 0.0259 | 0.0086 | 0.021 | 0.0203 | 0.014 | 0.0043 | 0.0117 |
| Improv. | 3.69% | 4.52% | 4.92% | 8.51% | 4.93% | 6.58% | 7.50% | 6.60% | 5.18% | 6.87% | 7.50% | 7.34% |
| VBPR+DVIB$_a$ | 0.0283 | 0.0188 | 0.0064 | 0.0154 | 0.0384 | 0.0261 | 0.0086 | 0.0214 | 0.0223 | 0.0146 | 0.0046 | 0.012 |
| Improv. | **4.43%** | **6.21%** | **4.92%** | **9.22%** | **5.21%** | **7.41%** | **7.50%** | **8.63%** | **15.54%** | **11.45%** | **15.00%** | **10.09%** |
| MMGCN[57] | 0.0258 | 0.0167 | 0.0058 | 0.0133 | 0.0252 | 0.0161 | 0.0054 | 0.0131 | 0.0143 | 0.0093 | 0.003 | 0.0075 |
| MMGCN+DVIB$_c$ | 0.0277 | 0.0181 | 0.006 | 0.0151 | 0.0296 | 0.0192 | 0.0067 | 0.0152 | 0.0219 | 0.0139 | 0.0045 | 0.0114 |
| Improv. | 7.36% | 8.38% | 3.45% | 13.53% | 17.46% | 19.25% | 24.07% | 16.03% | 53.15% | 49.46% | 50.00% | 52.00% |
| MMGCN+DVIB$_a$ | 0.0281 | 0.0184 | 0.0064 | 0.0151 | 0.030 | 0.020 | 0.0067 | 0.0164 | 0.0223 | 0.0144 | 0.0047 | 0.0116 |
| Improv. | **8.91%** | **10.18%** | **10.34%** | **13.53%** | **19.05%** | **24.22%** | **24.07%** | **25.19%** | **55.94%** | **54.84%** | **56.67%** | **54.67%** |
| GRCN[56] | 0.0348 | 0.0237 | 0.0077 | 0.0193 | 0.0384 | 0.0252 | 0.0085 | 0.0205 | 0.0281 | 0.018 | 0.0058 | 0.0147 |
| GRCN+DVIB$_c$ | 0.0357 | 0.0238 | 0.0078 | 0.0194 | 0.0399 | 0.0267 | 0.0089 | 0.0222 | 0.0299 | 0.0195 | 0.0062 | 0.0159 |
| Improv. | 2.59% | 0.42% | 1.30% | 0.52% | 3.91% | 5.95% | 4.71% | 8.29% | 6.41% | 8.33% | 6.90% | 8.16% |
| GRCN+DVIB$_a$ | 0.0364 | 0.0243 | 0.0082 | 0.0198 | 0.0412 | 0.0276 | 0.0091 | 0.0228 | 0.0302 | 0.0199 | 0.0063 | 0.0165 |
| Improv. | **4.60%** | **2.53%** | **6.49%** | **2.59%** | **7.29%** | **9.52%** | **7.06%** | **11.22%** | **7.47%** | **10.56%** | **8.62%** | **12.24%** |
| BM3[79] | 0.0345 | 0.0229 | 0.0076 | 0.0189 | 0.042 | 0.0278 | 0.0093 | 0.0224 | 0.0282 | 0.0184 | 0.0058 | 0.0152 |
| BM3+DVIB$_c$ | 0.0359 | 0.0242 | 0.0080 | 0.0196 | 0.0447 | 0.0297 | 0.0098 | 0.0241 | 0.0303 | 0.0198 | 0.0062 | 0.0158 |
| Improv. | 4.06% | 5.68% | 5.26% | 3.70% | 6.43% | 6.83% | 5.38% | 7.59% | 7.45% | 7.61% | 6.90% | 3.95% |
| BM3+DVIB$_a$ | 0.0365 | 0.0245 | 0.0082 | 0.0198 | 0.0448 | 0.0297 | 0.0099 | 0.0241 | 0.0309 | 0.0204 | 0.0063 | 0.0170 |
| Improv. | **5.80%** | **6.99%** | **7.89%** | **4.76%** | **6.67%** | **6.83%** | **6.45%** | **7.59%** | **9.57%** | **10.87%** | **8.62%** | **11.84%** |
| FREEDOM[77] | 0.0379 | 0.025 | 0.0083 | 0.0206 | 0.0454 | 0.0293 | 0.0098 | 0.0237 | 0.0395 | 0.0262 | 0.0082 | 0.0216 |
| FREEDOM+DVIB$_c$ | 0.0391 | 0.0257 | 0.0086 | 0.0212 | 0.0468 | 0.0312 | 0.0102 | 0.0255 | 0.0416 | 0.0272 | 0.0086 | 0.0224 |
| Improv. | 3.17% | 2.80% | 3.61% | 2.91% | 3.08% | 6.48% | 4.08% | 7.59% | 5.32% | 3.82% | 4.88% | 3.70% |
| FREEDOM+DVIB$_a$ | 0.0399 | 0.0264 | 0.0087 | 0.0218 | 0.047 | 0.0313 | 0.0103 | 0.0260 | 0.042 | 0.0278 | 0.0087 | 0.0229 |
| Improv. | **5.28%** | **5.60%** | **4.82%** | **5.83%** | **3.52%** | **6.83%** | **5.10%** | **9.70%** | **6.33%** | **6.11%** | **6.10%** | **6.02%** |
| Avg Improv. (DVIB$_c$) | 4.17% | 4.36% | 3.71% | 5.84% | 7.16% | 9.02% | 9.15% | 9.22% | 15.50% | 15.22% | 15.23% | 15.03% |
| Avg Improv. (DVIB$_a$) | 5.80% | 6.30% | 6.89% | 7.19% | 8.35% | 10.96% | 10.04% | 12.47% | 18.97% | 18.76% | 19.00% | 18.97% |

**Table 1: Top-5 recommendation performance of MRS with and without DVIB (DVIB$_c$ for constant noise and DVIB$_a$ for adaptive noise) on the Baby, Sports, and Clothing datasets. "Improv." indicates the relative improvement of DVIB over the baseline, while "Avg. Improv." represents the average enhancement across all datasets.**

recommendation performance. To compare with the adversarial training method AMR, Hits Ratio (HR) is adopted as an evaluation metric to maintain DVIB's consistency and comparability with the established benchmark in official result of AMR. All the above-mentioned metrics range from 0 to 1, the closer to 1 the better.

**Baselines.** We evaluate DVIB across a variety of recommendation models, including both multimodal and single-modal systems. Our multimodal baselines include Bayesian Personalized Ranking with matrix factorization (VBPR [15]), three graph neural networks (MMGCN [57], GRCN [56], FREEDOM [77]), and self-supervised learning methods (BM3 [79]). For single-modal models, we test DVIB on the self-supervised learning model SelfCF [78] and the graph-based LayerGCN [76]. To further assess DVIB's compatibility, we conduct experiments combining DVIB with robustness methods MG [67] and AMR [46], demonstrating that DVIB can work synergistically with other robustness methods for MRS.

## 4.2 Overall Performance

**Observation #1: DVIB significantly elevates the performance of various MRS.** We assess DVIB's effectiveness across five multimodal models using three datasets, namely Baby, Sports, and Clothing. As demonstrated in Table 1, the results consistently underscore the superiority of our approach, with DVIB outperforming baseline models across all metrics. The greatest improvement is observed in model MMGCN on dataset Clothing, where DVIB$_a$

achieves an impressive performance boost exceeding 50%. This notable improvement is attributed to DVIB's hidden-layer perturbations and self-distillation mechanism reducing its detrimental effect on recommendation accuracy. It also demonstrates empirically that DVIB does have potential to filter out redundant information, sharpening the model's focus on task-relevant features, and validating its adherence to VIB theory. Together, DVIB makes the model more robust to capture and leverage meaningful information, resulting in significantly improved performance across diverse metrics.

**Observation #2: Adaptive noise demonstrates clear superiority over constant noise.** This advantage is evident empirically as shown in Table 1. The dynamic nature of adaptive noise in DVIB$_a$ allows the model to automatically adjust the noise level based on the complexity and heterogeneity of the data, resulting in consistently better performance across various models and datasets. This capability enhances the model's robustness and generalization, proving its efficacy in MRS.

## 5 Analysis

To evaluate the performance and versatility of the DVIB framework, our analysis focuses on the following five research questions (RQs):
- RQ1: Does DVIB truly mitigate inherent noise in MRS?
- RQ2: Does DVIB truly handle redundancy in MRS?
- RQ3: Is DVIB compatible with other robust training methods?

- RQ4: How about the training efficiency of DVIB?
- RQ5: Can DVIB enhance single-modal recommender systems?

We mainly consider three MRS, including VBPR [15], MMGCN [57], BM3 [79] on the dataset Baby for the following analysis.

| Models | $\chi$ | Ori. | Noise | Decr. ↓ |
|---|---|---|---|---|
| MMGCN | | 0.0258 | 0.0245 | 5.04% |
| MMGCN+DVIB$_c$ | 0.01 | 0.0277 | 0.0264 | 4.69% |
| MMGCN+DVIB$_a$ | | 0.0281 | 0.0279 | **0.71%** |
| MMGCN | | 0.0258 | 0.0219 | 15.12% |
| MMGCN+DVIB$_c$ | 0.05 | 0.0277 | 0.0254 | 8.30% |
| MMGCN+DVIB$_a$ | | 0.0281 | 0.0277 | **1.42%** |

**Table 2: REC of MMGCN [57] under varying intensities $\chi$ of Gaussian noise injection into textual and visual modality on the dataset Baby. The noise intensity is represented by the standard deviation of Gaussian distribution. "Decr." denotes the relative comparison to the origin(Ori.) after noise injection.**

**Mitigating Inherent Noise and Redundant Information (RQ1 &RQ2).** We explicitly verify whether the proposed DVIB can truly mitigate the impact of inherent noise and redundant information by incorporating them into MRS.

(1) Inherent Noise. To model with learnable multimodal embedding layers, such as MMGCN [57], we simulate real-world information noise by injecting Gaussian noise of varying intensities into the embeddings. Specifically, the noise injection follows the formulation:

$$\text{Emb}_{\text{noisy}} = \text{Emb} + \chi \cdot \boldsymbol{\epsilon}, \quad \boldsymbol{\epsilon} \sim \mathcal{N}(0, \mathbf{I}) \qquad (24)$$

where Emb represents the original multimodal embeddings, $\chi$ is the noise intensity, and $\boldsymbol{\epsilon}$ is drawn from a standard normal distribution $\mathcal{N}(0, \mathbf{I})$. This noise injection mimics potential distortions or errors in real-world data, such as blurry item images or incorrect item textual information, like comments or descriptions. As shown in Table 2, DVIB consistently outperforms baseline models in both accuracy and robustness, demonstrating its resilience to noisy inputs. Additionally, DVIB$_a$, which employs adaptive noise, shows higher robustness compared to DVIB$_c$, further highlighting the advantage of the adaptive noise scale mechanism. Therefore, DVIB does mitigate inherent noise in multimodal data.

(2) Redundant Information. To further validate DVIB's ability to handle redundant information, we introduce irrelevant yet harmless redundancy into textual and visual modalities. In the textual modality (TR), we append 30-character random strings (drawn from a-z, A-Z, 1-9) to half of the samples. In the visual modality (VR), we augment the original 4096-dimensional visual features with 512-dimensional extra features consisting of absolute values from $\mathcal{N}(1, 9)$ combined with 512-dimensional extra features of zeros to simulate redundant visual features. The experiment results on MMGCN are shown in Table 3. We can see that DVIB effectively mitigates the negative impacts of both textual and visual redundancy (TR, VR). Even in scenarios where both types of redundancy are present (TR+VR), DVIB still maintains stable performance, providing a clear affirmative answer to its ability for handling redundancy in multimodal data.

| Models | Red. Type | Ori. | w/ Red. | Decr. ↓ |
|---|---|---|---|---|
| MMGCN | | 0.0260 | 0.0225 | 15.56% |
| MMGCN+DVIB$_c$ | TR+VR | 0.0277 | 0.0249 | 11.24% |
| MMGCN+DVIB$_a$ | | 0.0281 | 0.0275 | **2.18%** |
| MMGCN | | 0.0260 | 0.0228 | 14.04% |
| MMGCN+DVIB$_c$ | VR | 0.0277 | 0.0250 | 10.80% |
| MMGCN+DVIB$_a$ | | 0.0281 | 0.0277 | **1.44%** |
| MMGCN | | 0.0260 | 0.0244 | 6.56% |
| MMGCN+DVIB$_c$ | TR | 0.0277 | 0.0264 | 4.92% |
| MMGCN+DVIB$_a$ | | 0.0281 | 0.0279 | **0.72%** |

**Table 3: REC of MMGCN [57] with redundancy (Red.) in textual and visual modalities on the dataset Baby. "Decr." represents the relative decrease of models (w/ Red.) compared with the origin(Ori.). MMGCN is evaluated under three redundancy scenarios (Red. Modal): (1) TR+VR (both textual and visual redundancy), (2) VR (visual redundancy only), and (3) TR (textual redundancy only).**

| Models | Metrics | Ori. | AMR | AMR+DVIB$_c$ | AMR+DVIB$_a$ |
|---|---|---|---|---|---|
| VBPR [15] | HR | 0.1352 | 0.1395 | 0.1457 | **0.1460** |
| | NDCG | 0.1005 | 0.1027 | 0.1048 | **0.1050** |
| | | Ori. | MG | MG+DVIB$_c$ | MG+DVIB$_a$ |
| | REC | 0.0258 | 0.0263 | 0.0284 | **0.0288** |
| MMGCN [57] | NDCG | 0.0167 | 0.0172 | 0.0184 | **0.0187** |
| | PREC | 0.0058 | 0.0060 | 0.0062 | **0.0065** |
| | MAP | 0.0133 | 0.0138 | 0.0153 | **0.0154** |

**Table 4: Top-5 recommendation performance under different robust training methods AMR and MG for MRS with and without DVIB on the dataset Baby.**

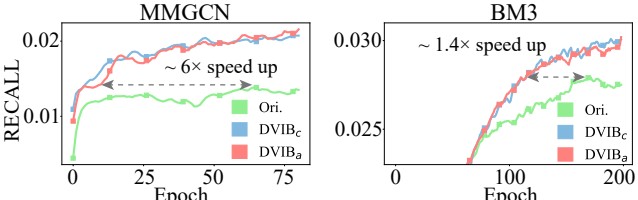

**Figure 3: REC of MMGCN [57] and BM3 [79] comparing the training efficiency. See more results in Appendix.**

**Compatibility with Robustness Methods (RQ3).** To further assess DVIB's versatility, we investigate its compatibility with existing robustness-enhancing methods including AMR [46] and MG [67]. The experiment results are shown in Table 4. When combining robustness methods with DVIB, the results show consistent improvements. These results highlight that DVIB is compatible with robustness-enhancing methods, further validating its flexibility and compatibility across different strategies for MRS robustness.

**Improving Training Efficiency (RQ4).** As shown in Fig. 3, we assess the training efficiency of DVIB by tracking its REC as training progresses. Following the training strategy in Zhou et al. [73], we set the maximum number of epochs to 1000, with an early stopping strategy to prevent overfitting. The results in Fig. 3 demonstrate that

| Models | Baby | | | | Sports | | | | Clothing | | | |
|---|---|---|---|---|---|---|---|---|---|---|---|---|
| | REC | NDCG | PREC | MAP | REC | NDCG | PREC | MAP | REC | NDCG | PREC | MAP |
| SelfCF [78] | 0.0345 | 0.0226 | 0.0074 | 0.0186 | 0.0421 | 0.0281 | 0.0090 | 0.0232 | 0.0271 | 0.0178 | 0.0057 | 0.0148 |
| SelfCF+DVIB$_c$ | 0.0354 | 0.0234 | 0.0078 | 0.0191 | 0.0428 | 0.029 | 0.0092 | 0.0242 | 0.0285 | 0.0186 | 0.0058 | 0.0155 |
| Improv. | 2.61% | 3.54% | 5.41% | 2.69% | 1.66% | 3.20% | 2.22% | 4.31% | 5.17% | 4.49% | 1.75% | 4.73% |
| SelfCF+DVIB$_a$ | 0.0357 | 0.0238 | 0.0078 | 0.0195 | 0.0449 | 0.03 | 0.0098 | 0.0249 | 0.0307 | 0.0205 | 0.0063 | 0.0173 |
| Improv. | **3.48%** | **5.31%** | **5.41%** | **4.84%** | **6.65%** | **6.76%** | **8.89%** | **7.33%** | **13.28%** | **15.17%** | **10.53%** | **16.89%** |
| LayerGCN [76] | 0.0337 | 0.0223 | 0.0072 | 0.0184 | 0.0394 | 0.0262 | 0.0083 | 0.0214 | 0.0256 | 0.0170 | 0.0051 | 0.014 |
| LayerGCN+DVIB$_c$ | 0.0338 | 0.0227 | 0.0076 | 0.0189 | 0.0399 | 0.0269 | 0.0089 | 0.0219 | 0.0263 | 0.0173 | 0.0054 | 0.0145 |
| Improv. | 0.30% | 1.79% | 5.56% | 2.72% | 1.27% | 2.67% | 7.23% | 2.34% | 2.73% | 1.76% | 5.88% | 3.57% |
| LayerGCN+DVIB$_a$ | 0.0345 | 0.023 | 0.0078 | 0.0192 | 0.0402 | 0.0269 | 0.0087 | 0.0222 | 0.0267 | 0.0175 | 0.0055 | 0.0145 |
| Improv. | **2.37%** | **3.14%** | **8.33%** | **4.35%** | **2.03%** | **2.67%** | **4.82%** | **3.74%** | **4.30%** | **2.94%** | **7.84%** | **3.57%** |
| Avg Improv. (DVIB$_c$) | 1.45% | 2.67% | 5.48% | 2.70% | 1.47% | 2.94% | 4.73% | 3.32% | 3.95% | 3.13% | 3.82% | 4.15% |
| Avg Improv. (DVIB$_a$) | **2.93%** | **4.22%** | **6.87%** | **4.59%** | **4.34%** | **4.72%** | **6.85%** | **5.53%** | **8.79%** | **9.05%** | **9.18%** | **10.23%** |

**Table 5: Top-5 recommendation performance of baseline single-modal recommender systems with and without DVIB (DVIB$_c$ for constant noise and DVIB$_a$ for adaptive noise) on the Baby, Sports, and Clothing datasets. "Improv." indicates the relative improvement of DVIB over the baseline, while "Avg. Improv." represents the average enhancement across all datasets. The DVIB is equally effective for single-modal recommender systems.**

the models enhanced by DVIB consistently achieve faster convergence speed, requiring fewer epochs to reach the same performance levels as their baseline counterparts, and finally achieving superior results. While DVIB introduces extra computational cost due to the extra paths (discussed in Section 7), the overall training cost remains acceptable given the substantial gains in convergence speed. Moreover, it is important to note that DVIB introduces no additional computational cost during inference, making it highly efficient in both training and deployment stages.

**Improvement on Single-modal Models (RQ5).** Single-modal systems can be viewed as a specific case of MRS, and in addition to its success in MRS, we further explore whether our DVIB can improve single-modal models with LayerGCN [76] and SelfCF [78]. As shown in Table 5, DVIB consistently delivers performance gains across multiple datasets, with the most striking improvement observed in SelfCF on the dataset Clothing, where the MAP score increased by an impressive 16.89% using the DVIB$_a$ approach. Although single-modal systems are less affected by the inherent noise and redundant information characteristic of multimodal data, our method still significantly enhances their robustness and generalization capabilities. These results illustrate that DVIB remains highly effective even in more straightforward, single-modal environments, which is still consistent with our theory in Section 3.2.

## 6 Ablation Study

We first investigate the effects of each component in our loss function, focusing on three specific terms in Eq. (8): general loss, self-distillation loss, and regularization term. Then, we evaluate the influence of various perturbation $\delta H$.

**(1) Effects of General Loss $\mathcal{L}_0^{\mathbf{DVIB}}$ and $\mathcal{L}_0$.** To understand the influence of general loss, we conduct the ablation study on $\mathcal{L}_0^{\mathrm{DVIB}}$ and $\mathcal{L}_0$ with BM3 on the dataset Baby. The results in Fig. 4 show that both $\mathcal{L}_0^{\mathrm{DVIB}}$ and $\mathcal{L}_0$ are essential for maintaining model performance. Removing any one of them will lead to a significant drop

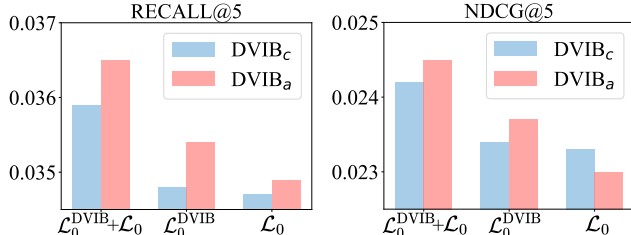

**Figure 4: REC and NDCG of BM3 [79] comparison for DVIB variations on the dataset baby, showing performance with different loss components: general loss $\mathcal{L}_0^{\mathbf{DVIB}}$ and $\mathcal{L}_0$.**

in performance. This highlights the importance of preserving both loss components to achieve optimal training outcomes.

**(2) Effects of Self-distillation Loss $\mathcal{L}_d^{\mathbf{DVIB}}$.** We first experiment with different consistency weight $\gamma_d$ in $\mathcal{L}_d^{\mathrm{DVIB}}$. Further analysis is conducted on using KL-divergence [27] versus JS-divergence [37] to compute the $\mathcal{L}_d^{\mathrm{DVIB}}$. The results in Fig. 5 show that it's crucial to adjust $\gamma_d$ to balance between original model feature and distilled knowledge, with certain weights allowing the model to maintain this balance more effectively. Additionally, the result in Table 6 shows that JS-divergence consistently outperforms KL-Divergence, as it better balances the asymmetric nature of KL-divergence, making it more suitable to handle complex and multimodal distributions in our proposed DVIB framework.

**(3) Effects of Regularization Term $\mathcal{L}_r^{\mathbf{DVIB}}$.** We perform two sets of experiments to assess the impact of $\mathcal{L}_r^{\mathrm{DVIB}}$. First, we vary the weight $\gamma_r \in [0, 0.001, 0.01, 0.1, 1]$ to find the optimal balance between regularization strength and model flexibility. Second, we investigate the influence of the parameter $a$ in $\mathcal{L}_r^{\mathrm{DVIB}}$, which controls the relationship between the learnable noise scale $\phi$ and a Gaussian distribution w.r.t. $a$. As observed in Fig. 6, while changes in weight $\gamma_r$ and the parameter $a$ do influence model performance, the impact is not overly drastic. The weight $\gamma_r$ of 0.001 consistently yields the best results, balancing regularization without constraining learning.

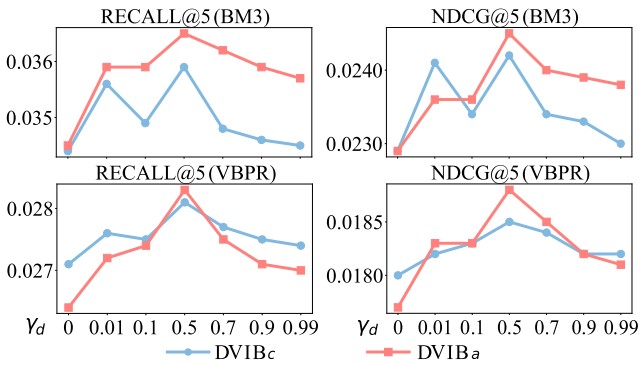

**Figure 5: NDCG and REC results with different values of consistency weight $\gamma_d$ in self-distillation loss Eq. (10).**

| Models | Divergence | REC | NDCG | PREC | MAP |
|---|---|---|---|---|---|
| | $KL(Z||Z_d)$ | 0.0346 | 0.0229 | 0.0076 | 0.0188 |
| BM3+DVIB$_c$ | $KL(Z_d||Z)$ | 0.0344 | 0.0227 | 0.0076 | 0.0187 |
| | JS | **0.0359** | **0.0242** | **0.008** | **0.0196** |
| | $KL(Z||Z_d)$ | 0.0346 | 0.0230 | 0.0076 | 0.0188 |
| BM3+DVIB$_a$ | $KL(Z_d||Z)$ | 0.0353 | 0.0232 | 0.0078 | 0.0189 |
| | JS | **0.0365** | **0.0245** | **0.0082** | **0.0198** |
| | $KL(Z||Z_d)$ | 0.0263 | 0.0171 | 0.0061 | 0.0148 |
| VBPR+DVIB$_c$ | $KL(Z_d||Z)$ | 0.0253 | 0.0167 | 0.0058 | 0.0147 |
| | JS | **0.0281** | **0.0185** | **0.0064** | **0.0153** |
| | $KL(Z||Z_d)$ | 0.0267 | 0.0177 | 0.0060 | 0.0143 |
| VBPR+DVIB$_a$ | $KL(Z_d||Z)$ | 0.0265 | 0.0177 | 0.0059 | 0.0143 |
| | JS | **0.0283** | **0.0188** | **0.0064** | **0.0154** |

**Table 6: Top-5 performance of BM3 [79] and VBPR [15] under different self-distillation methods. $KL(Z||Z_d)$ represents the KL-divergence [27] from the original hidden feature $Z$ to the perturbed hidden feature $Z_d$, $KL(Z_d||Z)$ represents the KL-divergence from the perturbed hidden feature to the original hidden feature, and JS represents the JS-divergence [37] between the two.**

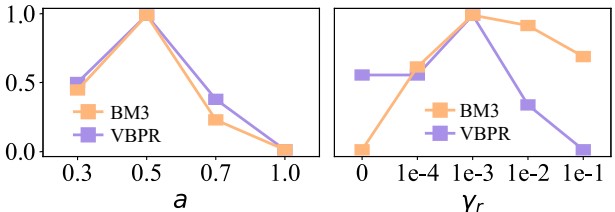

**Figure 6: Normalized REC of BM3 [79] and VBPR [15] on the dataset Baby with different values of parameter $a$ or regularization term weights ($\gamma_r$) as defined in Eq. (13).**

For the parameter $a$, a value of 0.5 emerges as optimal, indicating that it helps manage the noise adaptively without overly limiting the model's learning capacity. For further details on the data and analysis, please refer more results in the Appendix.

**(4) Different Ways of Perturbations in DVIB$_c$.** We evaluate the impact of different noise injection methods on model performance. Guided by the principles of VIB theory, our primary approach involves Gaussian noise, as defined in Eq. (11) and Eq. (12), which

introduces controlled perturbations with the variance dictating the degree of them. To assess the efficacy of this method, we compare it against two alternative strategies: dropout [44, 60] and uniform noise [4]. Dropout randomly removes a portion of network connections during training, introducing perturbations without the continuous properties of Gaussian noise. In contrast, uniform noise introduces randomness from a uniform distribution, lacking the central tendency around zero, which may lead to different perturbation characteristics.

The results in Table 7 clearly demonstrate that Gaussian noise is the most effective method for perturbations in the DVIB framework. This aligns with the theoretical expectations of VIB, where Gaussian noise provides controlled perturbations that regularize the model while preserving robustness. Dropout and uniform noise, lacking the structured variance and statistical properties of Gaussian noise, exhibit inferior performance in comparison.

| Models | Noise | REC | NDCG | PREC | MAP |
|---|---|---|---|---|---|
| | Gaussian | **0.0359** | **0.0242** | **0.008** | **0.0196** |
| BM3+DVIB$_c$ | Uniform | 0.0303 | 0.0207 | 0.0067 | 0.0174 |
| | Dropout | 0.0312 | 0.021 | 0.0069 | 0.0177 |
| | Gaussian | **0.0365** | **0.0245** | **0.0082** | **0.0198** |
| BM3+DVIB$_a$ | Uniform | 0.0347 | 0.0229 | 0.0077 | 0.0191 |
| | Dropout | 0.0292 | 0.0198 | 0.0062 | 0.016 |

**Table 7: Top-5 performance of BM3 [79] with different perturbation methods, including the effects of Gaussian, Uniform, and Dropout noise.**

## 7  Limiation

The proposed DVIB method also has some limitations. For instance,

(1) it requires extra computational cost during training. Fortunately, the extra DVIB path in Fig. 2 only adds about 10% to 20% more training time for various MRS. This path is not involved in computation during inference, resulting in no extra cost at that stage. Moreover, Fig. 3 shows that the training efficiency of our DVIB significantly outperforms that of the baseline. Therefore, the overall cost of DVIB is acceptable.

(2) The location of the perturbation $\delta H$. As mentioned in Section 2, different MRS models possess varying network structures, which may make it difficult to establish a unified rule for determining the exact layer where perturbation $\delta H$ should be applied. However, our experiments reveal that in most cases, the optimal position for introducing noise is after multimodal feature fusion, aligning with the theoretical expectations outlined in Section 3.2, and this location is typically straightforward to identify within the network structure. For more implementation details, please refer to the Appendix.

## 8  Conclusion

This paper introduces DVIB, simple-yet-effective framework that mitigates inherent noise and information redundancy risks in various MRS without altering the original network architecture. The effectiveness of DVIB is not only supported by the variational information bottleneck theory but also by extensive experiments across different datasets and model settings, representing a promising new paradigm in MRS.

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

## A    Related Works

**Multimodal Recommender Systems.** Traditional recommender systems [17, 65, 76, 78] primarily model user preferences and item properties by relying on implicit interaction data, such as ratings and clicks. However, these systems often face challenges like data sparsity[8] and cold start [43]. MRS address these limitations by incorporating diverse multimodal data, offering richer context for user preferences and item attributes, which leads to improved recommendation performance [34]. Early approaches predominantly relied on collaborative filtering [2, 38] and matrix factorization[15, 26]. More recently, deep learning methods, including graph neural networks [56, 57, 77], attention mechanisms [20, 24, 49, 68] and self-supervised learning [79] have been applied.

**Robustness of Recommender Systems.** Recent studies [10, 34, 53] have increasingly highlighted the vulnerability of RS to both noise and redundant information, which can significantly undermine the accuracy of recommendations. To improve the robustness of RS, much of the focus has been placed on adversarial training methods [3, 9, 11, 35, 40, 46, 58, 59, 62]. These approaches introduce controlled perturbations to input data or model parameters, simulating potential attack scenarios to enhance the model's resilience against noise. Additionally, some research [31, 33, 36, 49, 52, 54] attempts to design more complicated network structures incorporating methods, such as attention mechanisms [19, 21, 23, 29, 70, 71], to filter out redundant information to reduce the interference of redundant information on the model. However, existing methods typically address either noise or redundancy in isolation, lacking a comprehensive solution that tackles both issues simultaneously.

**Variational Information Bottleneck.** As data continues to scale and grow in complexity, balancing data compression with the retention of task-relevant information has become a critical challenge in machine learning. The Information Bottleneck (IB) method, introduced by Tishby et al. [50], addresses this by optimizing mutual information to compress input data while preserving the most critical features for a given task. To adapt IB to high-dimensional data, Alemi et al. [1] proposed the Variational Information Bottleneck (VIB), which integrates variational inference with stochastic gradient descent, making IB applicable to deep learning models. In recent years, VIB has also demonstrated notable effectiveness in RS [5, 30, 55].

## B    Implementation Details

We follow the standard settings for all baseline models. The Adam optimizer [25] is adopted for model optimization unless specified otherwise. Following the settings of some pioneer works in multimodal recommender systems [74], we conduct a grid search to determine the optimal weight of self-distillation loss $\gamma_d$, regularization term in $DVIB_a$ and noise scale for $DVIB_c$. Specifically, $\gamma_d$ is searched among values like 0.5, 0.1, 0.01, which align with common practices for self-distillation. For the regularization term in $DVIB_a$, it is calculated via the KL-divergence [27] between the model's distributions, and the hyperparameters $\alpha_0$ and $\beta_0$ can be constant. The general loss weight $\gamma_0$ is set to 1 as the regularization weight $\gamma_r$ set to 0.001 after testing a range of values where we found that 0.001 consistently provided better generalization. For the noise scale in $DVIB_c$, values like 0.001, 0.0005, and 0.0001 were tested. Meanwhile,

we set the hyper-parameter $a$ in $DVIB_a$ as 0.5. All model training and evaluation are performed on an NVIDIA RTX3090 GPU to ensure consistent and fair computations.

## C    The Details of Dataset

In this paper, we use the Amazon review dataset, a widely used benchmark in recommendation research, as the main dataset of our experimental evaluation. The dataset contains both item descriptions and corresponding images, enabling multimodal analysis. Following the approach of previous studies [74, 79], we use three distinct per-category datasets for comprehensive evaluation: Baby, Clothing, and Sports. These datasets vary in size and complexity, covering different item categories and providing a robust testbed for our proposed models. The statistics for these datasets, including their size and sparsity, are detailed in Table 8 as follows.

| Dataset | #Users | #Items | #Interactions | Sparsity |
| --- | --- | --- | --- | --- |
| Baby | 19,445 | 7,050 | 160,792 | 99.88% |
| Sports | 35,598 | 18,357 | 296,337 | 99.95% |
| Clothing | 39,387 | 23,033 | 278,677 | 99.97% |
| Pinterest | 3,226 | 4,998 | 9,844 | 99.94% |

**Table 8: Statistics of the experimental datasets. These datasets include both textual and visual features.**

The Pinterest dataset is chosen for our compatibility analysis with the AMR method [46] to ensure a fair comparison, as it is the same dataset used in the official AMR code for robustness experiments.

## D    Which Layer Should We Inject $\delta H$?

In our proposed DVIB framework, noise perturbations are introduced into the hidden layer of MRS to enhance model robustness. As discussed in Section 7, different MRS models have distinct network structures, making it challenging to establish a uniform rule for determining the exact layer at which noise should be injected. However, our experiments have shown that, in most cases, adding noise after multimodal feature fusion leads to significant performance improvements, which aligns well with our theoretical expectations.

Specifically, in early fusion network structures, noise can be introduced after feature fusion, giving the model ample time to learn from and adapt to the perturbations. Conversely, for models where fusion occurs later, if noise is injected after fusion, the later layers of the network may struggle to fully leverage the benefits of these perturbations, because the subsequent layers are not sufficiently expressive to fully capitalize on the noise perturbations. Therefore, introducing noise earlier, particularly before multimodal fusion, often yields better results.

In VBPR [15], textual and visual features are first concatenated to generate fused multimodal embeddings. Perturbations are then added to this fused item embedding before the original and noisy embeddings are passed to subsequent network layers for training. Similarly, in GRCN [56], perturbation is added after modality fusion, where the representations of the text and visual modalities are combined through GCN layers. In BM3 [79], as fusion is not explicitly performed, perturbation is injected separately into the textual and

visual features after generating the initial user and item embeddings. In FREEDOM [77], a multimodal adjacency matrix is used to create fused embeddings, and noise is introduced progressively based on the number of layers in the network.

It is worth noting that for models like MMGCN [57], where fusion occurs late in the network, the effectiveness of adding noise at a later stage is more limited. This is due to the insufficient representational power of the later layers, which prevents the model from fully utilizing the benefits of noise perturbations. As a result, for such models, it is more effective to inject noise earlier, ideally before modality fusion, to achieve optimal results.

In summary, the DVIB framework flexibly adapts to different MRS structures by introducing noise perturbations at appropriate hidden layers to improve performance. While noise is often injected after multimodal fusion in most cases, the optimal point of injection may vary depending on the specific model structure and the timing of feature fusion.

## E  Effects of Regularization Term

In this section, we present additional experimental results to illustrate the influence of various hyperparameter settings within the regularization terms. As shown in Table 9, the Top-5 performance peaks when the parameter $a$ is set to 0.5. This value of $a$ offers an optimal balance in the noise distribution between the model's learned embeddings and the standard Gaussian distribution, $N(0, a^2 I)$. When $a$ is too small, the noise added is insufficient, which hampers the model's ability to leverage the adaptive noise mechanism. Conversely, when $a$ is too large, the model over-prioritizes the noise, which can diminish the generalization capacity by skewing towards excessive noise rather than learning meaningful representations. Therefore, maintaining the right balance in $a$ is critical for maximizing the model's overall performance.

| Models | a | REC | NDCG | PRE | MAP |
|---|---|---|---|---|---|
| VBPR[15] | 0.3 | 0.0279 | 0.0188 | 0.0064 | 0.0154 |
|  | 0.5 | **0.0283** | **0.0188** | **0.0064** | **0.0154** |
|  | 0.7 | 0.0278 | 0.0187 | 0.0063 | 0.0153 |
|  | 1.0 | 0.0275 | 0.0186 | 0.0062 | 0.0153 |
| BM3[79] | 0.3 | 0.0360 | 0.0239 | 0.0079 | 0.0196 |
|  | 0.5 | **0.0365** | **0.0245** | **0.0082** | **0.0198** |
|  | 0.7 | 0.0358 | 0.0237 | 0.0078 | 0.0194 |
|  | 1 | 0.0356 | 0.0233 | 0.0077 | 0.0191 |

**Table 9: Top-5 performance of VBPR and BM3 on the Baby dataset with different values of parameter $a$ in Eq. (13).**

Similarly, Table 10 demonstrates the significance of selecting an appropriate regularization weight $\gamma_r$. Regularization helps prevent overfitting by ensuring the adaptive noise scale remains effective. Setting $\gamma_r$ too low can weaken the regularization effect, resulting in overfitting and diminished model robustness. Conversely, if $\gamma_r$ is too high, the model may become overly constrained, limiting its capacity to learn meaningful patterns from the data. Therefore, carefully tuning $\gamma_r$ is essential for achieving optimal regularization and, by extension, enhanced model stability and performance.

| Models | Metrics | 0 | 0.0001 | 0.001 | 0.01 | 0.1 |
|---|---|---|---|---|---|---|
| VBPR[15] | REC | 0.0279 | 0.0279 | **0.0283** | 0.0277 | 0.0274 |
|  | NDCG | 0.0186 | 0.0187 | **0.0188** | 0.0185 | 0.0183 |
|  | PREC | 0.0062 | 0.0063 | **0.0064** | 0.0063 | 0.0062 |
|  | MAP | 0.0152 | 0.0153 | **0.0154** | 0.0150 | 0.0148 |
| BM3[79] | REC | 0.0346 | 0.0354 | **0.0359** | 0.0358 | 0.0355 |
|  | NDCG | 0.0234 | 0.0235 | **0.0242** | 0.0240 | 0.0241 |
|  | PREC | 0.0077 | 0.0078 | **0.008** | 0.0080 | 0.0078 |
|  | MAP | 0.0191 | 0.0193 | **0.0196** | 0.0195 | 0.0194 |

**Table 10: Top-5 performance of VBPR and BM3 under different regularization term weights ($\gamma_r$) as defined in Eq. (13).**

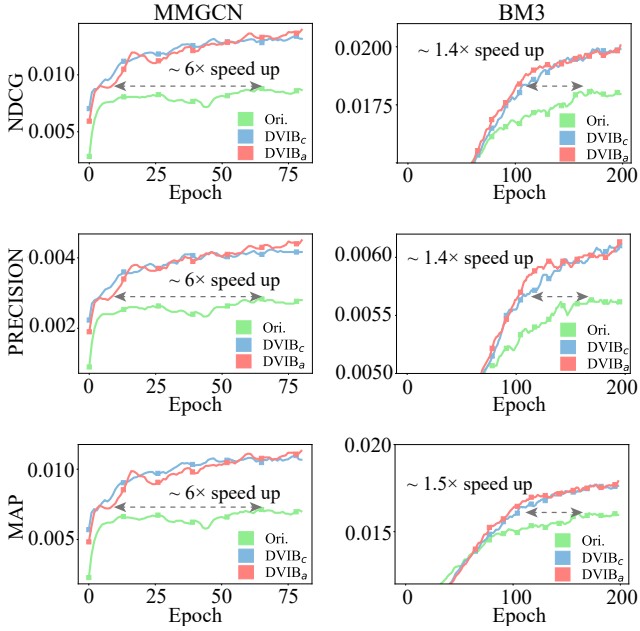

**Figure 7: Various metrics of MMGCN [57] and BM3 [79] with DVIB variations over different epochs.**

## F  Improving Training Efficiency

As discussed in Section 5, our DVIB framework not only accelerates training but also improves final performance. To further substantiate these claims, we provide additional metrics in Fig. 7, which highlights the significant gains in convergence speed as well as superior results.

In the MMGCN plots (left), DVIB achieves target NDCG, Precision, and MAP scores up to six times faster than the original model, reaching optimal performance within the first 25-50 epochs. Similarly, for the BM3 model (right), DVIB accelerates convergence by 1.4 to 1.5 times. Importantly, both models not only converge faster but also achieve higher final performance compared to the baselines, underscoring the dual benefit of speed and quality with DVIB.

## G  Different Ways of Perturbations in DVIB$_c$

Table 11 demonstrates the effects of different perturbation ways on the MMGCN [57] model, provided as supplementary results

to further support the findings in Section 6. Notably, Gaussian noise consistently outperforms Uniform and Dropout noise, which aligns with our Variational Information Bottleneck (VIB) theory [1], reinforcing the importance of controlled, Gaussian-distributed perturbations in optimizing model performance.

| Models | Noise | REC | NDCG | PRE | MAP |
|---|---|---|---|---|---|
| MMGCN+DVIB$_c$ | Gaussian | **0.0277** | **0.0181** | **0.006** | **0.0151** |
| | Uniform | 0.0271 | 0.0173 | 0.0058 | 0.0145 |
| | Dropout | 0.0272 | 0.0173 | 0.0055 | 0.0146 |
| MMGCN+DVIB$_a$ | Gaussian | **0.0281** | **0.0184** | **0.0064** | **0.0151** |
| | Uniform | 0.0256 | 0.0171 | 0.0058 | 0.0139 |
| | Dropout | 0.0276 | 0.0181 | 0.0061 | 0.0148 |

**Table 11: Top-5 performance of MMGCN [57] with different noise injection methods, including the effects of Gaussian, Uniform, and Dropout noise.**

