# OpenReview forum: "DVIB: Towards Robust Multimodal Recommender Systems via Variational Information Bottleneck Distillation"
_ACM.org/TheWebConf/2025/Conference — WWW 2025 Poster_

### Official Review · Reviewer_Pn1Y · 2024-11-03

**Novelty:** 4
**Technical Quality:** 4

**Review:**

The authors propose the DVIB framework, which reduces noise and redundancy in multimodal recommendation systems (MRS) without altering model architecture, enhancing robustness. Theoretical support links DVIB to the variational information bottleneck, and experiments show consistent performance improvements across datasets and compatibility with existing robust training methods.

I have only a general understanding of multimodal information systems and the information bottleneck theory. I can follow the DVIB method, but I don’t understand many details in the proof of DVIB. I am maintaining a neutral stance and have raised some questions. I will adjust my score based on the author’s responses to these questions and other reviewers' comments.

**Questions:**

- Does adding noise to the input $M_i$ or $X$ also implicitly exhibit an information bottleneck effect? If so, the information bottleneck effect may not serve as a compelling argument for the advantage of DVIB adding perturbations in the intermediate layer rather than in the input layer.

- Is this method of introducing noise and constraining the intermediate layer in DVIB proposed for the first time in this paper? Given the simplicity of the method, whether this is a novel contribution or an adaptation of existing work from other fields to the MRS domain is important for evaluating this work.

- Could the authors provide an intuitive explanation of how DVIB implicitly exhibits an information bottleneck effect?

- My last question is, isn't the process of optimizing the model itself equivalent to building information bottlenecks? Why can't they eliminate the so-called redundancy and noise?

**Reviewer Confidence:**

2: The reviewer is willing to defend the evaluation, but it is likely that the reviewer did not understand parts of the paper

**Scope:**

4: The work is relevant to the Web and to the track, and is of broad interest to the community

---

### Official Review · Reviewer_Zk5J · 2024-11-16

**Novelty:** 5
**Technical Quality:** 4

**Review:**

This text proposes a new framework that shifts the perturbations from the input layer to
the hidden layer and utilizes the feature self-distillation. This framework aims to
simultaneously mitigate the risks of the inherent noise and the redundant information in
multimodal recommender systems, without changing the original network architecture. A
theoretical analysis supports the rational design of the proposed framework, and
extensive experiments validate the effectiveness of the framework.

  This text should be weakly accepted.

Pros:
1. This text provides theoretical evidence to support the statement.
2. Extensive experiments show that the proposed framework significantly enhances the
performance of various models on various datasets.
3. The text is well-organized and presents the main idea clearly.


Cons:
1. The experiments do not include a performance comparison between the proposed framework and
the existing methods only concerning inherent noise and redundant information, respectively.

2. The ablation study is not complete. There is no ablation study to investigate why the idea
that moving the perturbations from the input layer to the hidden layer is effective.

  As for the grammatical mistake, in Figure 1 on Page 1, the "previous works for inherent noise"
should be "previous works for the inherent noise". The "previous works for redundant information"
should be "previous works for the redundant information".

**Questions:**

1. I wonder the performance comparison between the proposed framework and the existing methods only
concerning inherent noise and redundant information, respectively.

2. The main idea of the text is that moving the perturbations from the input layer to the hidden layer,
combined with feature self-distillation is effective. However, it seems that there is no experiment
demonstrating that the performance of the model with perturbations in the hidden layer surpasses that
in the input layer. Can you illustrate the motivation for moving the perturbations from the input layer
to the hidden layer?

3. There is no ablation study to compare the performance of the proposed framework with that of the
vanilla self-distillation training method to valiadate the effectiveness of adding of the
perturbations. Please explain the necessity of adding of the perturbations.

4. Can you explain the motivation for combining the proposed framework with the self-distillation
training method?

**Reviewer Confidence:**

3: The reviewer is confident but not certain that the evaluation is correct

**Scope:**

4: The work is relevant to the Web and to the track, and is of broad interest to the community

---

### Official Review · Reviewer_2PWX · 2024-11-27

**Novelty:** 5
**Technical Quality:** 4

**Review:**

This paper presents a novel framework, DVIB, which combines the Variational Information Bottleneck (VIB) with self-distillation to enhance the robustness of multimodal recommender systems (MRS). By shifting perturbations from the input to the hidden layer, DVIB effectively reduces noise and redundant information. Experiments demonstrate its performance improvement across various datasets and model settings.

Pros:

1.The DVIB framework combines VIB and self-distillation, presenting a novel approach to tackle robustness in MRS.

2.Empirical results show that DVIB enhances MRS performance across various datasets and model settings.

3.The paper is mostly clear and well-organized.

Cons:

1.No code is disclosed, making it difficult to reproduce.

2.The paper lacks crucial parameter sensitivity analysis.

3.Experiments on more datasets can further enhance the persuasive power of the method.

**Questions:**

The proposed method is straightforward, supported by theoretical proofs, and its effectiveness is demonstrated through experiments. However, the ablation study only explores the loss function and perturbations. As shown in Figure 1 (1), it would be valuable to investigate the effects of simply replacing Input Perturbation with Hidden Perturbation or substituting Robust Training with Self-Distillation Training. How would these individual changes impact the results?

**Reviewer Confidence:**

3: The reviewer is confident but not certain that the evaluation is correct

**Scope:**

4: The work is relevant to the Web and to the track, and is of broad interest to the community

---

### Official Review · Reviewer_Dahk · 2024-12-02

**Novelty:** 4
**Technical Quality:** 4

**Review:**

Strengths
1. Innovative Framework: The proposed DVIB framework integrates variational information bottleneck techniques with hidden-layer perturbations and self-distillation. This approach is both novel and theoretically sound, offering a unified solution for addressing noise and redundancy issues in multimodal recommender systems.
2. Theoretical Justification: The authors provide robust theoretical evidence for the efficacy of DVIB. The mathematical formulations and proofs convincingly demonstrate how the approach reduces noise and redundant information while improving feature extraction quality.
3. Comprehensive Evaluation: The experiments span multiple datasets and baseline models, showing consistent performance improvements. The adaptability of DVIB with both constant and adaptive noise further highlights its practical utility.

Weaknesses
1. Lack of Real-World Case Studies: While the theoretical and experimental settings are rigorous, the absence of real-world deployment scenarios limits the practical demonstration of the proposed framework's effectiveness.
2. Computational Cost Analysis: Although the paper mentions no additional inference cost, the training efficiency gains need more detailed exploration, particularly concerning scalability for larger datasets.
3. Limited Discussion on Generalization: The adaptability of DVIB to other domains beyond multimodal recommender systems, such as general machine learning tasks, is not addressed.

**Questions:**

1.Integration with Existing Methods: The paper shows compatibility with other robustness-enhancing methods (e.g., AMR, MG). Could you elaborate on the challenges faced during such integrations and the potential limitations of DVIB when applied alongside complex existing pipelines?
2. Adaptive Noise Scalability: The adaptive noise mechanism demonstrates superior performance. However, how does this method scale with increasingly large multimodal datasets or higher-dimensional features? Are there any observed bottlenecks?
3. Evaluation Metrics Beyond Top-5: The experiments primarily focus on Top-5 metrics. Can the authors provide insights into the framework's performance for broader rankings or less popular recommendations, which might reveal different aspects of robustness and generalization?

**Reviewer Confidence:**

3: The reviewer is confident but not certain that the evaluation is correct

**Scope:**

3: The work is somewhat relevant to the Web and to the track, and is of narrow interest to a sub-community

---

### Official Review · Reviewer_t5WE · 2024-12-02

**Novelty:** 5
**Technical Quality:** 4

**Review:**

This paper proposes a framework named DVIB (Variational Information Bottleneck Distillation), which achieves model-agnostic mitigation of the inherent noise and redundancy risks in multi-modal recommender systems simultaneously.

**Pros:**
1. This paper is well-written, and the proposed method is easy to understand.
2. The proposed method is well-rounded with a solid theoretical basis.
3. The conducted experiments seem to demonstrate the effectiveness of the proposed method.

**Cons:**

See questions.

**Questions:**

1. Regarding the main experiments shown in Table 1, it would be beneficial to add several other model-agnostic frameworks for denoising and removing redundancy as baselines to compare with DVIB.

2. Regarding the noise and redundancy experiments shown in Table 2 and Table 3, it would be better to conduct experiments on other base models, not just MMGCN.

**Reviewer Confidence:**

2: The reviewer is willing to defend the evaluation, but it is likely that the reviewer did not understand parts of the paper

**Scope:**

3: The work is somewhat relevant to the Web and to the track, and is of narrow interest to a sub-community